# Fracture-Related Infections: Current Status and Perspectives from the International Society of Antimicrobial Chemotherapy

**DOI:** 10.3390/antibiotics14111095

**Published:** 2025-11-01

**Authors:** Julie Lourtet-Hascoët, Eric Bonnet, Anna Maria Spera, Tiziana Ascione, Monica Chan, Silvano Esposito, Pasquale Pagliano, Antonia Scobie, Serhat Ünal, Gérard Giordano, Kordo Saeed

**Affiliations:** 1Laboratoire Infection et Inflammation, INSERM L2i, UFR des Sciences de la Santé Simone Veil, 78180 Montigny Le Bretonneux, France; 2Centre Regional d’Antibiotherapie, Service des Maladies Infectieuses et Tropicales, CHU de Toulouse, 31300 Toulouse, France; bonnet.e@chu-toulouse.fr; 3Department of Infectious Diseases, University of Salerno, 84121 Salerno, Italy; aspera@unisa.it (A.M.S.); silvanoesposito@libero.it (S.E.); 4Department of Infectious Diseases, AORN dei Colli, 80100 Naples, Italy; tizianascione@hotmail.com (T.A.); ppagliano@libero.it (P.P.); 5National Centre for Infectious Diseases, Department of Infectious Diseases, Tan Tock Seng Hospital, Singapore 308442, Singapore; monica.chan@nhghealth.com.sg; 6Infectious Diseases and Microbiology, Royal Free London NHS Foundation Trust, London NW3 2QG, UK; antonia.scobie@nhs.net; 7Department of Infectious Diseases, Faculty of Medicine, Hacettepe University, 06230 Ankara, Turkey; sunal@hacettepe.edu.tr; 8Orthopedic Surgery Department, Joseph Ducuing Hospital, 31300 Toulouse, France; ortho.giordano@gmail.com; 9Microbiology Innovation and Research Unit (MIRU), Microbiology Department, Southampton University Hospitals NHS Foundation Trust, Southampton SO16 6YD, UK; kordosaeed@nhs.net; 10School of Medicine, University of Southampton, Southampton SO17 1BJ, UK

**Keywords:** fracture-related infection, orthopedic infection, osteosynthesis, open limb, internal fixation, osteomyelitis, antibiotic treatment, surgery

## Abstract

Fracture-related infections (FRIs) represent a significant complication in orthopedic trauma care, often leading to delayed bone healing, prolonged hospital stays, and increased patient morbidity. Pathogenesis involves microbial contamination during injury or surgery, compounded by patient-related risk factors such as diabetes, smoking, or immunosuppression. Diagnosis of FRI relies on a combination of clinical, radiological, and microbiological criteria. Common signs include persistent pain, swelling, erythema, purulent discharge, and non-union of the fracture. FRIs are classified based on the timing of infection onset into acute, delayed, and chronic forms, each requiring tailored management strategies. Treatment generally involves aggressive surgical debridement, possible hardware removal or retention, and targeted antibiotic therapy. In cases of severe tissue loss, reconstructive procedures may be necessary to restore bone and soft tissue integrity. Treatment strategies include early administration of prophylactic antibiotics, meticulous surgical technique, and timely soft tissue coverage in open fractures. A multidisciplinary approach involving orthopedic surgeons, infectious disease specialists, and microbiologists is essential for successful management. Early recognition and appropriate intervention are crucial to improving outcomes and minimizing long-term disability in patients with fracture-related infections.

## 1. Introduction

Two million fracture fixation devices are inserted annually in the United States [1]. Fracture-related infections (FRIs) are a serious complication and one of the most frequently encountered bone and joint infections, with annual projected infections of 100,000 [1].

The overall FRI rate is estimated at 5%. FRI incidence ranges from 1 to 2% after internal fixation of closed fractures, to more than 30% after open fracture fixation [2]. This incidence is rising, representing an economic burden with substantial healthcare costs. Complex multidisciplinary management is necessary with prolonged hospitalization. In 2019, 178 million new fractures occurred worldwide. Economic impact is significant with total inpatient cost evaluated over a decade, up to AUD 157,000 in Australia. Infected trauma cases cost three times more than uninfected cases [3].

The average cost of treatment is estimated between AUD 15,000 and 25,000 [1]. Early or acute infections less than 2 weeks after implantation are the most common and are associated with virulent organisms such as *Staphylococcus aureus* and aerobic Gram-negative bacilli [2]. Delayed infections occur 4–10 weeks after surgery and late infections more than 10 weeks after surgery [2]. Delayed and late infections are harder to eradicate due to the presence of biofilm that shields bacteria from immunity and antibiotics. This persistence complicates eradication and favors chronicity. In contrast to prosthetic joint infections, hematogenous infections involving fracture fixation devices is uncommon [2].

Diagnostics can be challenging in cases of subtle or delayed symptoms, requiring complex surgeries associated with prolonged antimicrobial therapy. Finally, a multidisciplinary approach is crucial for optimal care and follow-up in these complex cases.

Some gaps persist in our understanding of FRIs, leading to difficulties in their management. The mechanism of biofilm formation, especially in vivo, remains unclear, limiting the efficiency of treatments. Diagnostic tools show limitations in identifying low-grade infections delay their management. Moreover, polymicrobial infections with evolving resistance profiles are frequent in open fractures, making the choice of adapted antibiotics challenging.

In this review, several ‘hot topics’ on FRIs were selected and reviewed by members of the Bone and Skin & Soft Tissue Infections Working Group of the International Society of Antimicrobial Chemotherapy (ISAC). This group includes international scientists, microbiology and infectious disease clinicians, and academics whose aim is to advance the education and science of infection management. This paper aims to summarize an updated comprehensive review of current evidence, providing a summary of the various aspects of FRI: epidemiology, diagnosis, clinical signs, treatment concepts, and prophylaxis. Opinions from experts on surgery, infectious diseases, and microbiology are highlighted as well as areas for future study and research.

## 2. Methodology

Data was collected on the definition of infectious complications after fracture fixation used in each study. Study selection was performed first by reviewing titles and abstracts for relevance, and then by getting the full texts of relevant articles. Full-text articles were reviewed in a narrative synthesis. Experts in bone and joint infections, including infectious disease specialists, surgeons, and microbiologists, examined data concerning fracture-related infections. Infections associated with fractures were retrieved by searching Embase, Cochrane, Google Scholar, Medline (OvidSP), PubMed. Articles in English published after 1980 were considered. Keywords included fracture-related infection, orthopedic infection, osteosynthesis, open limb, internal fixation, osteomyelitis, antibiotic treatment, injury, and surgical implant. In addition, when scientific evidence was lacking, recommendations were based on expert opinion.

## 3. Classification

Although loosely used to describe a clinical or subclinical infection of a fracture following surgical fixation, the specific definition of an FRI was developed by expert consensus from the Arbeitsgemeinschaft fur Osteosynthesefragen (AO) Foundation and European Bone and Joint Infection Society (EBJIS) [4]. The definition was further refined in 2020 following a second consensus meeting from this international expert group [4,5]. Two levels of certainty around diagnostic features are detailed in Table 1 [4,6].

## 4. Risk Factors

External and internal risk factors are detailed in Table 2.

Early administration of parenteral antibiotics after open trauma is warranted to reduce the risk of soft tissue infection. This should not be delayed and ideally should be given within one hour after trauma [12].

A meta-analysis of 1106 patients with open fractures found that antibiotics had a protective effect against early infection compared with no antibiotics or placebo, with a risk ratio of 0.43 (95% confidence interval of 0.29–0.65) and absolute risk reduction of 0.07 (95% CI 0.03–0.10) [13]. A 5-day course of antibiotics had no greater benefit than a 1-day course of antibiotics for prevention of infection [14].

## 5. Epidemiology and Causative Agents

In open fractures, infection predominantly arises from the environmental contamination of the injury site. For closed fractures, bacterial inoculation may occur during surgery. Hematogenous infection originating from distant sites is uncommon [2]. Polymicrobial infections are frequently reported, with rates up to 45%, higher than in periprosthetic joint infections (PJIs), which are around 10% [15,16].

A broad range of microorganisms are described in these infections. *Staphylococcus aureus* is the pathogen mainly described; coagulase-negative staphylococci, enterococci, and Gram-negative bacteria (GNB) are also commonly isolated [16,17]. Other pathogens such as *P. aeruginosa*, anaerobes, *C. acnes*, mycobacteria, and fungi may also be observed [16,17].

*S. aureus* is the main pathogen involved in FRIs, regardless of time of onset, reported in 27% to 60% of cases [16,17] followed by *S. epidermidis* and other coagulase negative staphylococci.

Regarding antimicrobial resistances of *S. aureus*, methicillin-resistant *Staphylococcus aureus* (MRSA) trends indicate a decline in the last 10 years in Europe and North America.

In chronic osteomyelitis, the rate of MRSA also showed a reduction over a ten-year period, from 30.8% to 11.4% [18]. The rate of methicillin-resistant *Staphylococcus* spp. is high, especially *Staphylococcus epidermidis*, showing 60% to 79.8% of beta-lactam resistances [19,20].

Linezolid, glycopeptides and daptomycin are antibiotics active against Gram-positive bacteria. They can be used to treat infections caused by resistant organisms such as *E. faecium*, MRSA, and CoNS, but require MIC testing. Daptomycin is also used in PJIs caused by resistant organisms. Animal models have shown an improved success rate when combined with rifampicin [21].

Gram-negative bacteria are frequent pathogens in FRIs, particularly in patients with open fractures. They are often associated with environmental contamination, with incidence rates up to 40%, which is higher than in other fractures (28%) or PJIs (14%) [15,22]. Polymicrobial infection due to Enterobacterales is more frequent in early FRI [20]. Risk factors for Gram-negative infections have been analyzed by Konda et al. in a retrospective review of 3360 patients [23], and the authors identified initial external fixation (*p* = 0.038), soft tissue coverage of an open fracture site (*p* = 0.039), and a lower serum albumin level at the onset of infection (*p* = 0.005) as major drivers of FRIs.

Gitajn et al. conducted a retrospective review of 423 patients requiring surgery for deep surgical site infection and found a co-occurrence of infections due to *Enterobacter*, *Enterococcus* and *Pseudomonas aeruginosa* at least 75% of time. It is probably related to high failure rate of infection management after trauma [24].

International guidelines for the treatment of MDR GNB infections do not cover bone and joint infections, and only case reports or small case series are available [25,26]. However, recommendations for the treatment of severe MDR GNB infections are available [25,27]. The rate of MDR GNB amounts to 16.4–47% of FRIs. In a British study, 13.6% of all GNB strains were expressing ESBL, ampC, or cabapenemase [16]. In a recent Belgian study, approximatively a quarter of GNB strains were resistant to piperacillin–tazobactam [20]. In low resources countries, the rate of GNB MDR can be significantly higher, with piperacillin–tazobactam resistance rate reported up to 43% [28,29].

## 6. Clinical Signs and Evolution

The clinical presentation of FRI can vary from a mild type to an acute or systemic one. Local and systemic signs of inflammation are the most common signs of FRI [21]. According to the classification, confirmatory or suggestive criteria are described [4].

A systematic review of randomized controlled trials performed by Metsemakers et al. in 2018 provides information about the variability of the clinical presentation of FRIs [30]; purulent drainage appears to be the most common clinical sign confirming FRI. Osteomyelitis and sepsis represent the most severe complication of FRI, often requiring surgical debridement and implant removal [4,30].

## 7. Diagnosis

Hematology/biochemistry:

Serum inflammatory markers used in the diagnosis of FRI include C-reactive protein (CRP), leukocyte count, and erythrocyte sedimentation rate [31]. These markers typically show a non-specific increase during the acute phase of injury, surgery, or other inflammatory condition. Conversely, levels of these markers may remain normal in cases of chronic or late-onset infections. CRP is widely used and has a reported sensitivity and specificity rate of 67% (95% CI, 52–80%) and 61% (95% CI, 47–99%), respectively. Interleukin-6 (IL-6), D-dimer, interferon-alpha, and procalcitonin have also been studied on FRI, but their predictive values are limited [31].

Microbiology:

The goal of microbiological diagnosis is to detect pathogens on FRI and confirm infection. Specimens for diagnosis include deep tissue, bone, and implant samples, as well as fluid samples from around the fracture or defect and surrounding tissue [32]. Three to five deep samples should be collected with clean, separate instruments and transferred rapidly to the laboratory. Standard culture techniques under aerobic and anaerobic conditions are the reference methods for diagnosis [18]. Techniques using liquid enrichment broths and blood culture bottles are useful [33]. Due to the presence of biofilm in bone infections, sonication or vortexing can help dislodge biofilm bacteria from implant surfaces. Sonication sensitivity rates range from 65 to 95%, with specificity between 50 and 97% [34].

Molecular diagnosis can also contribute to microbiological identification. Multiplex PCR or 16S-PCR can be useful for pathogen detection. Targeted metagenomics or shotgun metagenomics can detect a broader range of bacterial species and antimicrobial resistance genes. However, routine application is limited by cost considerations. Diagnosis is confirmed when at least two deep samples are positive for the same pathogenic bacteria. A single positive culture suggests infection and should be confirmed, except when bacteria like *S. aureus*, Gram-negative bacteria, and beta-haemolytic streptococci [18,32,33] are detected.

Histopathology:

Few studies report the accuracy of histopathology on FRI. However, studies recommend analyzing deep tissue samples from the inflammation area with at least 10 high power fields (HPF). A high correlation has been demonstrated between aseptic non-union and the absence of polymorphonuclear neutrophils (PMNs). Conversely, a threshold of 5 PMNs per HPF is necessary to confirm an FRI diagnosis [35].

Imaging:

Conventional radiography remains a first-line imaging modality due to its wide availability and ability to detect fracture lines, alignment, and initial signs of infection [2]. However, its sensitivity is limited in the early stages of infection.

Magnetic resonance imaging (MRI) has become invaluable in assessing FRIs, providing detailed images of both bone and surrounding soft tissues. MRI with contrast enhancement can differentiate infection and other causes of inflammation such as post-surgical changes or aseptic loosening. MRI is highly sensitive, often above 90%, for the detection of soft tissue infections, but its specificity may be affected by the presence of inflammation and postoperative changes [36]. Computed tomography (CT) is particularly useful when metallic implants are present [37]. Nuclear medicine techniques, such as Technetium-99m-labeled bone scintigraphy, play a role in the early detection of skeletal infections [38]. FDG-PET/CT and WBC scintigraphy with SPECT/CT both demonstrate good diagnostic accuracy for FRI [39].

Several studies have evaluated the effectiveness of FDG-PET/CT in diagnosing FRIs, and the results are promising. FDG-PET/CT is particularly useful in complex cases where conventional imaging is inconclusive. It may guide surgical decision making and antibiotic therapy by pinpointing active infection sites [40]. Once an FRI is detected, imaging continues to play a role in assessing the response to treatment. MRI allows for monitoring of soft tissue changes and the resolution of abscesses. Repeated imaging may be necessary to evaluate the effectiveness of antibiotic therapy and the need for further surgical intervention [41]. Ultrasound serves as a helpful adjunct in the evaluation of soft tissue collections and can guide aspiration or drainage procedures in real time. There is a need for optimized imaging protocols and better-defined diagnostic criteria to improve the accuracy and reliability of FRI diagnosis [36].

## 8. Fracture-Related Infections: Surgical Treatment

A multidisciplinary approach incorporates surgical intervention, systemic and local antibiotics, and close postoperative management [42,43]. Despite a notable absence of standardized, evidence-based guidelines for treating FRI, treatment protocols are proposed [32,42,43]. The diagnostic algorithm for a suspected fracture-related infection (FRI), which uses the updated definition criteria of the FRI Consensus Group, is helpful to determine the optimal treatment planning [5].

The keys points are fracture stabilization, debridement, irrigation with normal saline, dead space management, soft-tissues coverage, and host optimization (ceasing smoking, diabetes balance, vascular status, and nutrition).

To manage osteosynthesis, three concepts are discussed, depending on fracture’s healing status [32,43,44]: implant retention (DAIR), implant removal, or implant exchange.

Early and aggressive surgical debridement is essential in removing all infected and necrotic tissues while preserving viable bone and soft tissue. Complementary irrigation should be performed using normal saline at low pressure. These steps reduce the bacterial load, control local infection, and optimize conditions for subsequent reconstructive procedures [43].

A staged approach is often necessary for the most severe infections. The first stage focuses on debridement and temporary stabilization, often using external fixation. Once the infection is controlled, the second stage involves definitive reconstruction, such as bone grafting or internal fixation, to restore anatomy and function [4,43]. In some cases, a third (intermediate) stage is added, to make flaps or skin grafts.

Proper soft tissue management consists of techniques such as negative pressure wound therapy (NPWT) to promote wound healing and reduce infection risk [45].

If significant soft tissue loss occurs, free or rotational flap coverage using muscle or fascia-cutaneous flaps may be necessary to ensure adequate vascular supply and healing [46].

There is a need for standardized patient outcome measures focusing specifically on FRI. The Patient-Reported Outcomes Measurement Information System (PROMIS) is suggested as a preferred tool [43].

## 9. Fracture-Related Infections: Local Antimicrobial Treatment

The use of local antibiotic delivery systems, such as antibiotic-loaded cement beads or spacers, is an effective complement to systemic antibiotics. These devices deliver high concentrations of antibiotics directly to the site of infection, which can improve bacterial eradication and promote bone healing [46]. It is also necessary to limit dead spaces and show efficacy against resistant organisms and biofilms [47,48]. Prophylactic application of local antibiotics, including implant coatings, significantly reduces the risk of FRI compared to systemic prophylaxis alone [35,48].

Morgenstern et al.’s meta-analysis suggests a risk reduction on FRI of 11.9% if additional local antibiotics are given prophylactically for open limb fractures [35]. Vancomycin and gentamicin are frequently used due to their broad-spectrum activity against common pathogens in FRI and low resistance rates. Ciprofloxacin, tobramycin, and cephalosporins are also used based on microbial patterns [47,48,49].

Various delivery vehicles are used in the local antimicrobial treatment of FRI, including beads, spacers, gels, and coatings. Polymethylmethacrylate (PMMA) cement is mostly used [47,49,50]. These devices can be impregnated with antibiotics and inserted during surgery to deliver drugs over an extended period. However, PMMA has limitations such as the need for removal and limited antibiotic compatibility [50]. Antibiotic-loaded autograft could appear as an optimal solution of dead space management, enhanced biology, and infection control [51].

New absorbable bio composites, such as polylactic glycolic acid (PLGA) and a gentamicin-loaded calcium-sulfate/hydroxyapatite, are highly effective in the treatment of chronic osteomyelitis [52].

Bioglass materials have been explored, showing promise in treating FRI by providing antibacterial effects and supporting bone regeneration. Nanoglass paste containing silicate glass particles with calcium and copper has shown significant antibacterial effects against *E. coli* and *S. aureus*, promoting bone healing and angiogenesis in infected hard tissues [53].

Bioglass-based antibiotic-releasing putty demonstrated effective local antibiotic release, complete bacterial eradication, and support for bone regeneration in osteomyelitis models [54].

Innovations in disrupting biofilms include the use of enzymes like lysostaphin that degrade biofilm matrices. Surface coatings with antimicrobial peptides and antibiotic-loaded materials show promise in reducing bacterial attachment and biofilm formation on implants [48]. Nanoparticles offer a promising innovation for local antimicrobial delivery due to their ability to penetrate biofilms and deliver drugs at the nanoscale [55]. Silver nanoparticles, due to their inherent antimicrobial properties, are being extensively researched [56]. Finally, utilizing genomic and proteomic data can help in tailoring local antimicrobial therapies.

## 10. Antibiotic Prophylaxis

Uniformized recommendations are not available, but recent guidelines propose strategies for prophylaxis in FRIs [47,55,56,57]. Identifying high-risk patients, such as those with comorbidities or open fractures, is essential to tailoring prophylactic measures.

Early administration of antibiotics is crucial to prevent bacterial colonization and infection [58].

For less severe injuries like closed fractures, antibiotics targeting Gram-positive bacteria are recommended. First generation cephalosporin, like cefazolin, is sufficient and limited to a single dose before surgery [32]. According to the Gustilo classification, patients with type I or II injuries have antibiotics continued during 24 h. Extended prophylaxis may be proposed in case of Gustilo–Anderson type III injuries for up to 72 h or until wound closure [32,59].

There is no consensus for systematic prophylactic Gram-negative antibiotic coverage but, in severe injuries, broad-spectrum antibiotics targeting Gram-positive and Gram-negative bacteria are mostly recommended [60,61]. The need for prophylaxis was assessed in a retrospective review conducted by Ferreira et al. among 118 patients who underwent surgical treatment for FRI [62]. Perioperative anti-Gram-negative antibiotic prophylaxis for patients with external fixation requiring soft tissue coverage, at risk for malnutrition and prolonged hospitalization was also recommended by Konda et al. [23].

However, in a recent study, prophylactic Gram-negative antibiotic coverage seemed not to be justified for type III Gustilo–Anderson open fractures, according to a recent study by Suzuki et al. They demonstrated no significant difference in FRIs between both groups [63]. A systematic survey comparing antibiotic prophylaxis management in Gustilo–Anderson type II and III open fractures showed over 90% broad coverage against Gram-positive and Gram-negative bacteria [59].

Some studies suggest that additional local antibiotics could decrease bacteria colonization leading to a reduced risk of FRI [64].

## 11. Empiric Treatment of Fracture-Related Infections

Treatment of fracture-related infections (FRIs) is a difficult task requiring an integrated approach considering both surgical procedure and a prolonged course of antibiotic therapy to obtain an infection-free fracture consolidation. Broad-spectrum antibiotics can be administered after intraoperative samples are collected.

Administration of antibiotics follows local guidance based on epidemiological data, which demonstrates that most cases are sustained by Gram-positive bacteria such as *Staphylococcus aureus* or *S. epidermidis*, with a high rate of methicillin resistance [19,65]. Gram-negative bacteria, including Enterobacterales or *Pseudomonas* spp., account for less than one-quarter of the cases, but this percentage can be higher in cases of open fractures. For these reasons, an empirical approach should ideally propose an active treatment against Gram-positive bacteria including MSSA, MRSA, streptococci, and anaerobes using not only a combination of amoxicillin–clavulanic acid or ampicillin–sulbactam, but also an agent active against methicillin-resistant staphylococci-like vancomycin (or another glyco/lipo-peptide or an oxazolidinone). In the study from the international consensus group on FRI, ampicillin–sulbactam alone were proposed in cases of sterile cultures [21].

The choice of anti-Gram-negative agent could depend on the local epidemiology, with a choice of a broad-spectrum beta-lactam (piperacillin–tazobactam, a third- or fourth-generation cephalosporin, or carbapenems). A recent study from a major British trauma center reported that 96.3% of infection episodes would have been covered by the empirical systemic antibiotic choice of teicoplanin and meropenem [16].

A retrospective study on 117 patients with FRI highlighted that about 40% of the patients were infected with *S. aureus*. Instead, Gram-negative bacteria were identified in those with a delay (2–10 weeks after implant placement) with the highest frequency (23%). A combination of glycopeptides and fluoroquinolones, meropenem–vancomycin, or gentamicin–vancomycin could be a valuable choice [33,65]. A retrospective study analyzing a case-series of 75 patients with FRI receiving empirical antimicrobial treatment highlighted that vancomycin associated with ceftazidime showed the lowest mismatch rate [66]. Due to the increase of MDR Gram-negative bacteria, antibiotic therapy should be adapted if the microbial documentation shows Enterobacterales producing extended-spectrum beta-lactamases (ESBL), carbapenem-resistant Enterobacterales (CRE), or Carbapenem-resistant *P. aeruginosa* (CRPA) (Table 3).

In cases of ESBM-producing bacteria, the preferred antimicrobial agents for treating infections due to ESBL-producing Enterobacterales are the carbapenems imipenem, meropenem, and ertapenem [25]. There is sufficient data, including PK/PD studies, to support the use of carbapenems in bone infections [67].

In ESCMID and IDSA guidelines for the treatment of ESBL-producing Enterobacterales infections, it is suggested to not use cephamycins and cefepime [25]. However, cefepime has high bone concentration and can be considered a good option for treatment of an AmpC (non BLSE)-producing Enterobacterales FRI. Temocillin can be another option to treat an ESBL-E FRI, although it has also not been licensed for BJI. In a recent study by Lahouati et al., among 17 patients treated with temocillin for various BJI, 8 of the 12 patients (66.7%) who completed at least 3 months of follow-up, were considered cured [68].

In the case of carbapenemase-producing bacteria, the most clinically important carbapenemases are the following: KPC (*Klebsiella pneumoniae* carbapenemases), NDM-1 (New Delhi metallo-beta-lactamase (MBL)), and OXA-48.

**Table 3 antibiotics-14-01095-t003:** Antibiotics and combinations treating resistant Gram-negative bacteria [69].

	ESBL	KPC	MBL	AmpC	OXA-48	*P. aeruginosa* MDR/XDR	*Acinetobacter* MDR/XDR	*S. maltophilia **
Aztreonam/avibactam								
Cefepime/enmetazobactam								
Cefepime/taniborbactam								
Cefepime/zidebactam								
Cefiderocol								
Ceftaroline/avibactam								
Ceftolozane/tazobactam								
Ceftazidime/avibactam								
Imipenem/relebactam								
Meropenem/nacubactam								
Meropenem/vaborbactam								

Green: antimicrobial activity; Red: no activity; Yellow: partial activity; Grey: not available; ESBL: extended-spectrum β lactamase; KPC: *K. pneumoniae* carbapenemase; MBL: metallo-β-lactamase; AmpC: cephalosporinase; OXA-48: oxacillinase-48; MDR: multi-drug resistant; XDR: extended-drug resistant. * Resistant strains to TMP/SMX or levofloxacin.

Preferred agents active on carbapenemase-producing Enterobacterales (CPE) infections include ceftazidime–avibactam for KPC and OXA-48 but also for MBL (in combination with aztreonam), and imipenem–relebactam and meropenem–vaborbactam for KPC. Cefiderocol is also an alternative. Data is rarely reported on the bone tissues concentration of these recent combinations.

## 12. Recent or Alternative Therapeutic Options

Ceftazidime-avibactam: Good diffusion of bone tissues has been demonstrated for ceftazidime in some studies [67], but no data is available for avibactam. Ceftazidime–avibactam has shown to be effective in rabbit models of *E. coli* OXA-48 and *K. pneumoniae* KPC osteomyelitis [70]. In clinical practice, there are only some case reports or small cases series for MDR GNB bone and joint infections, including mainly *P. aeruginosa* infection and ESBL-producing Enterobacterales infection [71,72].

Imipenem–relebactam: For imipenem, the ratio of concentrations in the bone to those in the bone marrow has been estimated at 34.6% [73]. There are some case reports showing promising results for the treatment of bone infection by imipenem–relebactam [74].

Meropenem–vaborbactam. For meropenem the overall concentrations in bone are quite good, estimated at 10.5 g/mL [67]. There are few clinical reports on the use of meropenem-–vaborbactam for treating CRE bone and joint infections (BJI) [75].

Cefiderocol: In one case report, the value of cefiderocol in bone was satisfactory (13.9 g/mL). In clinical practice, data on cefiderocol efficacy for treating bone infection is scarce. In the study by Giacobbe et al., which comprised 22 patients with MBL-Enterobacterales infections, only 4 patients among 200 patients included in the analysis had BJI. The global cure rate for the 22 patients was 77% [76].

The need to combine another class of antibiotics with beta-lactams is not clear. However, in cases of multi-drug resistance, when there is only one beta-lactam option with a MIC close to the susceptibility threshold, adding an antimicrobial agent such as colistin, fosfomycin, aminoglycosides (for a short period), or tigecycline is an option. When resistance affects all beta-lactams, including the newest ones, using a combination of these antibiotics may be the only option.

Fosfomycin exhibits satisfactory bone concentration (ratio bone/plasma = 0.46) [77]. The activities of fosfomycin, tigecycline, colistin, and gentamicin were evaluated against extended-spectrum-β-lactamase-producing Escherichia coli in a foreign-body infection model: Fosfomycin in monotherapy was more effective than tigecycline, gentamicin, and colistin, but the combinations with fosfomycin improved the results. Specifically, the combination of fosfomycin and colistin achieved sterilization of the box in 67% of the cases [62,78].

Colistin may be the only antibiotic active against MDR GBN; however it is not recommended as monotherapy, especially in BJI. In a recent multicenter study of adults with BJI-resistant GNB treated with beta-lactams plus colistin, the overall rate of MDR/XDR GNB infections was 27/44 (61%) and the cure rate was 82%. In an international retrospective study including 131 drug-resistant Gram-negative prosthetic joint infections, it was shown that success rates did not differ for colistin versus non-colistin in XDR cases [69].

CRE isolates are often susceptible to at least one aminoglycoside (e.g., gentamicin, tobramycin, amikacin, and plazomicin). Plazomicin retains activity against CRE isolates that are resistant to all other aminoglycosides. But clinical data using plazomicin to treat systemic infections due to CRE is limited [79]. Pharmacokinetic studies demonstrated that aminoglycosides effectively penetrate bone and joint tissues, despite their hydrophilicity [67]. However, due to their toxicity profile, aminoglycosides are reserved for specific clinical indications, and prolonged administration is generally not recommended.

The activity of tetracycline derivatives, including new drugs such tigecycline and eravacycline, is not altered by carbapenemases. However, data on the use of these new tetracyclines in the treatment of BJI is lacking.

## 13. Oral Treatment

Switching to oral antibiotics should be considered after culture and sensitivity tests are available to complete the planned course of treatment. In any case, quinolones use has been questioned either as monotherapy or as an alternative agent to beta-lactams because they can rapidly develop resistance [32,80,81].

No study tailored to FRI treatment can assess the usefulness of anti-biofilm agents such as rifampin in these cases. In clinical practice guidelines, the IV route is recommended for probabilistic treatment and as the first line for targeted treatment for BJI. When active antibiotics have good oral bioavailability, the need for prolonged IV administration is not obvious [82].

The main antibiotics available by oral route that have been used to treat BJI are amoxicillin, amoxicillin–clavulanate, cephalexin, flucloxacillin, fluoroquinolones, clindamycin, TMP–SMZ, tetracyclines, oxazolidinones (linezolid, tedizolid), rifampicin (and other rifamycins), and metronidazole. However, prolonged use of linezolid beyond 14 days may lead to serious adverse effects, including bone marrow suppression (e.g., thrombocytopenia, anemia) and peripheral or optic neuropathy. Regular monitoring of hematologic parameters and neurological status is recommended during extended therapy. It is important to note that ciprofloxacin and levofloxacin are the only oral antibiotics effective against *Pseudomonas* spp. [21,25,43].

Oral antibiotics used for the treatment of Gram-positive- and Gram-negative-related FRI, their bioavailability, average dosages, and valuable spectrum are summarized in Table 4.

Some Enterobacterales resistant to carbapenemases and fluoroquinolones are susceptible to TMP–SMZ; resistance to TMP–SMX is common among CRE isolates [83]. Despite clinical data regarding treatment of CRΕ infections by TMP–SMZ being scarce, it can be a reasonable option to treat susceptible CRE FRI.

### 13.1. When to Start Oral Treatment

The time for switching from the IV to the oral route is still being debated. In the study “OVIVA”, which included 1054 patients with bone and joint infections with or without osteosynthetic material, partial oral therapy was not inferior to complete IV treatment [84]. Other results summarized in the narrative review by Besal et al. confirmed the non-inferiority of partial oral treatment, which involved an oral relay after a maximum IV therapy duration of 14 days [83]. In a study examining periprosthetic joint infections caused by Gram-positive cocci, Coehlo et al. found that the remission rates were similar compared with a group of patients who received empirical intravenous antibiotics post-surgery [81]. Since most oral antibiotics used for treating FRI have high bioavailability, we recommend administering them as soon as bacterial results become available. This strategy shortens hospital stays and lowers the incidence of complications [85].

### 13.2. Suppressive Oral Antimicrobial Treatment

Suppressive oral treatment is an option when surgical treatment has been suboptimal, and when the residual infection is still active after a complete course of curative antibiotic therapy.

The most prescribed antibiotics include tetracyclines, cotrimoxazole, and clindamycin. Other antibiotics that may also be utilized include amoxicillin, cephalexin, and fluoroquinolones. Quinolones should only be used for chronic infections caused by Gram-negative bacteria resistant to oral beta-lactams or cotrimoxazole. New long-acting IV-administered antibiotics like dalbavancin appear to be appealing alternatives to oral antibiotics for the suppressive treatment of bone joint infections [87]. The duration of suppressive treatment is not clearly defined. It usually varies from 6 to 12 months and may even last longer in some cases, sometimes for life [80].

## 14. Fracture-Related Infections: Follow-Up

The duration of antibiotic treatment in FRIs is not exactly defined and varies depending on the presence of hardware. Opinions are based on clinical experience and practices with other bone infections, such as joint prostheses, or osteitis in diabetic patients. If the hardware is completely removed without replacement, we suggest a treatment duration of 4 to 6 weeks. After conservative surgical treatment or when hardware is left in place or replaced, the recommended duration is the same as for joint prosthesis infections, which is 3 months [88]. In cases of amputation, a 3- to 4-week course of antibiotics may be recommended for partial amputation (residual osteitis) and 5-day course in cases of complete excision of the osteomyelitic zone without skin and soft tissue infection [89].

A complete follow-up should be conducted by a multidisciplinary team including clinical, surgical, biological, and imaging evaluation. Clinical follow-up consists of a regular assessment of symptoms monitoring, functional evaluation, and antibiotic therapy during checkup appointments. During surgical follow-up, procedure monitoring and surveillance of complications is performed due to regular evaluations. Inflammation markers like CRP/ESR are monitored at baseline and then for up to 6 months. Imaging tests like X-rays are carried out after surgery and periodically for up to 12 months.

## 15. Innovative Approaches for Fracture-Related Infections

Novel strategies for treating infections related to fractures have shown promise recently. Continuous local antibiotic perfusion (CLAP) therapy has emerged as a potential strategy for eliminating infections, particularly in refractory situations. Additionally, research on bacteriophage therapy, which utilizes viruses, that specifically infect bacteria leading to lysis of the bacterial cell wall, has the potential to improve treatment outcomes [90]. It has emerged as a promising approach for treating FRIs [91].

Within a historical cohort of 1307 patients in Poland, 41 FRI cases were described as having received administration of phages with success rates of approximately 60% [92]. A review by Suh et al. estimated an overall efficacy rate of 85%. This included implant retention amongst PJI cases [93]. A case series by Onsea et al. outlined both treatment and prevention of four polymicrobial FRIs with administration of intraoperative phages [94].

At present, most reported cases involve use of non-GMP phages under local compassionate use schemes. There is significant heterogeneity between routes of administration, phage dose concentrations, dosing intervals, and treatment durations. Most successful cases appear to involve a combination of phage therapy alongside antibiotics [93,95,96].

Another approach with potential use for FRI therapy is vaccines. The use of vaccines represents promises for prevention and therapeutic intervention. Adjuvants like β-glucans have been investigated to enhance vaccine efficacy against pathogens, suggesting potential applications for FRI therapy [97].

A randomized clinical trial evaluated a recombinant five-antigen Staphylococcus aureus vaccine (rFSAV) in patients undergoing elective surgery for closed fractures. Furthermore, vaccines hold potential for preventing FRI by stimulating a specific humoral immune response, reducing infection risks following musculoskeletal trauma [98]. The determination of the ideal phage dose remains a subject of debate. The challenges and key components of vaccines warrant further investigation.

## 16. Conclusions

Fracture-related infections remain a significant challenge in trauma care, because they are influenced by external and host risk factors. Prompt interventions, including early antibiotic administration, are crucial in reducing infection risks. The incidence of infection is significantly higher after open fractures than after closed fractures. Early administration of empirical antimicrobial treatment and prompt surgical management are essential to preventing infection, especially in cases of open fractures. Postoperative infection management includes early surgical and medical treatment targeting Gram-positive, Gram-negative, and anaerobic bacteria. For chronic infections, the timing of antibiotic administration depends on surgical strategies regarding the preservation of foreign bodies: Perioperative (after microbiological sampling) probabilistic treatment is necessary if hardware is replaced or remains in place, while delayed treatment (waiting for microbiological results) should be used when all foreign bodies are removed. In cases of multi-drug-resistant bacteria, the role of new beta-lactams needs confirmation, as does the need for antibiotic combination. Except when amputation is performed, the duration of antimicrobial therapy varies from 4 to 12 months. Oral treatment can be initiated early and the need for prolonged IV antibiotics remains unestablished. Local antibiotic delivery systems, bioglass materials, and phage therapy are new options that can reduce the need for or the duration of antibiotic treatment. Recent therapeutic advancements, such as antibacterial-coated implants, localized antibiotic delivery systems, and immunomodulatory therapies, offer promising solutions for prevention and treatment. These innovations highlight future perspectives for improving recovery rates and reducing complications in patients with FRI. Fracture-related infections remain a significant challenge in trauma care, influenced by external and host risk factors. Prompt interventions, including early antibiotic administration, are crucial to reducing infection risks. Some gaps in diagnosis and treatment show the need for optimized strategies. Recent therapeutic advancements, such as antibacterial-coated implants, localized antibiotic delivery systems, and immunomodulatory therapies, offer promising solutions for prevention and treatment. These innovations highlight future perspectives for improving recovery rates and financial impact on healthcare systems.

## Figures and Tables

**Table 1 antibiotics-14-01095-t001:** Classification of fracture-related infections.

	Confirmatory Criteria	Suggestive Criteria
Clinical	Sinus tract or fistula Wound breakdown with communication to bone or implant Purulent drainage from wound or presence of pus during surgery	Any of the following: Pain—new onset, increasing over time Local redness Local swelling Local temperatures increased Fever ≥ 38.3 °C
Biomarkers		Elevated ESR Elevated WBC Elevated CRP
Radiological		Bone lysis Implant loosening Non-union Sequestration Presence of periosteal bone formation
Microbiological	Phenotypically indistinguishable organisms identified from two or more separate deep tissue specimens	Pathogen identified from a single deep tissue specimen
Histology	Visible microorganisms on histological analysis Presence of >5 neutrophils per high-powered field	

**Table 2 antibiotics-14-01095-t002:** FRI risk factors [7,8,9,10,11].

**External (Environmental)** **Risk Factors**	- Type, site, and degree of injury - Virulence of pathogens - Geographical location and seasonal factors - Severity of fracture (e.g., Gustilo and Anderson classification) - Degree of wound contamination - Extent of soft tissue injury - Location and type of fracture (e.g., lower limb fracture, pathological fracture) - Polytrauma, blast, penetrating, and combat injuries - Multiple reconstruction surgeries - Pulsatile lavage of fracture - Blood transfusion and splenectomy - Farmyard injuries - Delay of prophylactic antibiotics > 6 h
**Internal (Host)** **Risk Factors**	- Advanced age (>80 years) - Male gender - Smoking - Diabetes mellitus - Presence of malignancy - Respiratory disease - Systemic immunodeficiency - Poor nutritional status - Peripheral vascular disease - Immunosuppressive conditions or medications

**Table 4 antibiotics-14-01095-t004:** Oral antibiotics used for the treatment of FRI [43,78,83,84,85,86].

	Bioavailability				Proposed Dosages	Useful Spectrum of Activity
	60–70%	70–80%	80–90%	90–100%		
Amoxicillin			X		1000 mg/6 h to 2000 mg/8 h	*Streptococcus* spp. (safely administered if MIC less than 0.25 mg/L), *E. faecalis*, *E. coli* (wild strains), *Pasteurella*, *C. acnes*, *Finegoldia* sp.
Amoxicillin–clavulanic acid	X *				1000 mg/6 to 8 h	Idem amoxicillin + methicillin-sensitive (MS) staphylococci, penicillinase-producing *E. coli*, various anaerobes. ***
Cephalexin [52,77,78]			X		1000 mg/8 h	MS staphylococci, *Streptococcus* spp., *E. coli* (wild strains), *C. acnes*.
Flucloxacillin	X *				1000 mg/6 h	*S. pyogenes*, MS staphylococci.
Clindamycin				X	600 mg/8 h	MS (and MR) staphylococci, *S. pyogenes*, *Streptococcus* spp., *C. acnes* and other anaerobes, *B. cereus*.
Levofloxacin				X	500–750 mg/24 h **	MS *S. aureus* (in combination with another antibiotic mainly rifampicin), CNS, *Streptococcus* spp., *Enterobacterales*, *P. aeruginosa*, *B. cereus.*
Ciprofloxacin		X			500–750 mg/12 h	MS *S. aureus*, CNS, *Streptococcus* spp., *Enterobacterales*, *P. aeruginosa*, *B. cereus.*
Moxifloxacin			X		400 mg/24 h	MS *S. aureus*, CNS, *Streptococcus* spp., anaerobes.
TMP–SMZ				X	1600/320 mg/12 h	Staphylococci, Enterobacterales.
Tetracyclines (Doxycycline, Minocycline)				X	100 mg/12 h	Staphylococci, *C. acnes* (some staphylococci strains may be resistant to doxycycline and susceptible to minocyclin).
Oxazolidinones (Linezolid, Tedizolid)				X	600 mg/12 h 200 mg/24 h	Staphylococci., *Streptococcus* spp., *Enterococcus* spp., *C. acnes* and other Gram-positive anaerobes.
Rifampicin				X	10 mg/kg/d	Staphylococci, *Streptococcus* spp. (anti biofilm activity. Used only in combination).

* The oral bioavailability of clavulanic acid and flucloxacillin is nearly 60%. ** 500 mg for BGN and 750 mg for staphylococci. *** Amoxicillin–clavulanic acid can be an option for some polymicrobial infections.

## Data Availability

Not applicable.

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
