# Peer review of "Fracture-Related Infections: Current Status and Perspectives from the International Society of Antimicrobial Chemotherapy"

_antibiotics, 2025, doi:10.3390/antibiotics14111095_

Round 1

Reviewer 1 Report

Comments and Suggestions for Authors

Introduction

The author should specify certain aspects of interest in the objective statement, not only general. (read the objective statement of Depypere, 2020; He, 2023)

Page 2 lines 60-63: This paper is an in-depth review of the current literature, providing a summary of the various aspects of FRI and expert opinions and insights from the authors’ own experience, highlighting areas for future study and research.

Reference:

Depypere M, Morgenstern M, Kuehl R, Senneville E, Moriarty TF, Obremskey WT, Zimmerli W, Trampuz A, Lagrou K, Metsemakers WJ. Pathogenesis and management of fracture-related infection. Clin Microbiol Infect. 2020 May;26(5):572-578. doi: 10.1016/j.cmi.2019.08.006. Epub 2019 Aug 22. PMID: 31446152.

He SY, Yu B, Jiang N. Current Concepts of Fracture-Related Infection. Int J Clin Pract. 2023 Apr 25;2023:4839701. doi: 10.1155/2023/4839701. PMID: 37153693; PMCID: PMC10154639.

Results and Discussion

The author should reorganize the subtitle in chronological order: classification – risk factor – epidemiology and causative agent – clinical sign and evolution – diagnosis – fracture related infections: surgical treatment – fracture related infections: local antimicrobial treatment – antibiotic prophylaxis – empiric treatment of fracture related infections – oral treatment – innovative approaches for fracture-related infections.

Conclusions

The author should give conclusion due to the study’s objective.

Comments on the Quality of English Language

Moderate

Author Response

Reviewer 1:

-Introduction

The author should specify certain aspects of interest in the objective statement, not only general. (Read the objective statement of Depypere, 2020; He, 2023)Page 2 lines 60-63: This paper is an in-depth review of the current literature, providing a summary of the various aspects of FRI and expert opinions and insights from the authors’ own experience, highlighting areas for future study and research.

Reference: Depypere M, Morgenstern M, Kuehl R, Senneville E, Moriarty TF, Obremskey WT, Zimmerli W, Trampuz A, Lagrou K, Metsemakers WJ. Pathogenesis and management of fracture-related infection. Clin Microbiol Infect. 2020 May;26(5):572-578. doi: 10.1016/j.cmi.2019.08.006. Epub 2019 Aug 22. PMID: 31446152.

He SY, Yu B, Jiang N. Current Concepts of Fracture-Related Infection. Int J Clin Pract. 2023 Apr 25; 2023:4839701. doi: 10.1155/2023/4839701. PMID: 37153693; PMCID: PMC10154639.

Thank you for this comment on the objective statement. The manuscript was revised accordingly.

“This paper aims to summarize an updated comprehensive review of current evidence, providing a summary of the various aspects of FRI: epidemiology, diagnosis, clinical signs, treatment concepts and prophylaxis. Opinions from experts on surgery, infectious diseases and microbiology are highlighted as well as areas for future study and research.”

-Results and Discussion

The author should reorganize the subtitle in chronological order: classification – risk factor – epidemiology and causative agent – clinical sign and evolution – diagnosis – fracture related infections: surgical treatment – fracture related infections: local antimicrobial treatment – antibiotic prophylaxis – empiric treatment of fracture related infections – oral treatment – innovative approaches for fracture-related infections.

We changed the organization of subtitles according to your comments.

-Conclusions

The author should give conclusion due to the study’s objective.

Thanks for adding this point. We modified the manuscript according to this comment.

“Fracture-related infections remain a significant challenge in trauma care, influenced by external and host risk factors. Prompt interventions, including early antibiotic administration, are crucial to reducing infection risks. The incidence of infection is significantly higher after open fractures than after closed fractures. Early administration of empirical antimicrobial treatment and prompt surgical management are essential to prevent infection, especially in cases of open fractures. Postoperative infection management includes early surgical and medical treatment targeting Gram-positive, Gram-negative, and anaerobic bacteria. For chronic infections, the timing of antibiotics depends on surgical strategies regarding the preservation of foreign bodies: perioperative (after microbiological sampling) probabilistic treatment is necessary if hardware is replaced or remains in place, while delayed treatment (waiting for microbiological results) should be used when all foreign bodies are removed. In cases of multi-drug resistant bacteria, the role of new beta-lactams needs confirmation, as does the need for antibiotic combination. Except when amputation is performed, the duration of antimicrobial therapy varies from 4 to 12 months. Oral treatment can be initiated early and the need for prolonged IV antibiotics remains unestablished. Local antibiotic delivery systems, bioglass materials, and phage therapy are new options that can reduce the need for or the duration of antibiotic treatment. Recent therapeutic advancements, such as antibacterial-coated implants, localized antibiotic delivery systems, and immunomodulatory therapies, offer promising solutions for prevention and treatment. These innovations highlight future perspectives for improving recovery rates and reducing complications in patients with FRI. Fracture-related infections remain a significant challenge in trauma care, influenced by external and host risk factors. Prompt interventions, including early antibiotic administration, are crucial to reducing infection risks. Some gaps in diagnostic and treatment show the need for optimized strategies. Recent therapeutic advancements, such as antibacterial-coated implants, localized antibiotic delivery systems, and immunomodulatory therapies, offer promising solutions for prevention and treatment. These innovations highlight future perspectives for improving recovery rates and financial impact on healthcare system.

Reviewer 2 Report

Comments and Suggestions for Authors

The authors present a review and opinion article on behalf of the International Society of Antimicrobial Chemotherapy regarding the management of fracture-related infections. The topic is relevant to clinical practice, as it presents a grey area.

General comments:

Authors should state an opinion or evidence-based recommendation about the length of the antibiotic treatment that should be administered in different scenarios, such as fracture consolidation with hardware removed, fracture consolidation, and hardware not removed, as well as the clinical and laboratory follow-up to determine clinical cure (ESR, CRP, imaging, etc.).

Line 186: “FDG- 186 PET/CT and WBC scintigraphy with SPECT/CT both demonstrate good diagnostic accuracy for FRI”

Comment: As FDG-PET/CT becomes more available in various settings, its diagnostic performance should be specified in the text.

Table 3: 9th row: Comment: Although some of the new β-lactam/beta-lactamase antibiotics have shown activity against S. maltophilia, authors should clarify that, depending on susceptibility, the treatment of choice remains TMP/SMX or levofloxacin.

Lines 302-303: Pharmacokinetic studies demonstrated that aminoglycosides effectively penetrate bone and joint tissues, despite their hydrophilicity”

Comment: Authors should emphasize that treatment with aminoglycosides for extended periods (as required for osteoarticular infections) is limited by its toxicity and may not be routinely recommended.

Lines 315-318: The main antibiotics available by oral route that have been used to treat BJI infections are amoxicillin, amoxicillin-clavulanate, cephalexin, flucloxacillin, fluoroquinolones, clindamycin, TMP-SMZ, tetracyclines, oxazolidinones (linezolid, tedizolid), rifampicin (and other rifamycins), and metronidazole.”

Comment: A warning about bone marrow toxicity and neuropathy caused by linezolid when used for more than 2 weeks should be included.

Lines 330-331: “Fosfomycin exhibits satisfactory bone con- 330

centration (ratio bone/plasma = 0.46)”

Comment: The fosfomycin dose should be specified or recommended for osteoarticular infections, as many oral dosages vary depending on the site of infection.

Minor comments:

Line 228: Pseudomonas sp.

Comment: Please italicize to Pseudomonas spp. (and add a second p at the end).

Table 3: line 10: Imipeneme/relebactam should be corrected to Imipenem/relebactam

Line 320: Pseudomonas sp.

Comment: Please italicize to Pseudomonas spp. (and add a second p at the end).

Table 4: Comment: Many of the words contained in the table, besides the microorganisms’ names, are incorrectly italicized.

Author Response

The authors present a review and opinion article on behalf of the International Society of Antimicrobial Chemotherapy regarding the management of fracture-related infections. The topic is relevant to clinical practice, as it presents a grey area. General comments: Authors should state an opinion or evidence-based recommendation about the length of the antibiotic treatment that should be administered in different scenarios, such as fracture consolidation with hardware removed, fracture consolidation, and hardware not removed, as well as the clinical and laboratory follow-up to determine clinical cure (ESR, CRP, imaging, etc.).

The duration of antibiotic treatment in FRI is not exactly defined and varies depending on the presence of hardware. Opinions are based on clinical experience and practices with other bone infections, such as joint prostheses, or osteitis in diabetic patients. If the hardware is completely removed without replacement, we suggest a treatment duration of 4 to 6 weeks. After conservative surgical treatment or when hardware is left in place or replaced, the recommended duration is the same as for joint prosthesis infections, which is 3 months.

In cases of amputation, we recommend a 3 to 4-week course of antibiotics for partial amputation (residual osteitis) and 5 days in cases of complete excision of the osteomyelitic zone without skin and soft tissue infection.

A complete follow up may be conducted by a multidisciplinary team including clinical, surgical, biological and imaging evaluation. Clinical follow up consists of regular assessment of symptoms monitoring, functional evaluation, and antibiotic therapy during checkup appointments.

During surgical follow up, procedure monitoring, surveillance complications are performed due to regular evaluations. Inflammation markers like CRP/ESR are monitored at baseline and then up to 6 months. Imaging like X-rays are carried out after surgery and periodically up to 12 months.

-Line 186: “FDG- 186 PET/CT and WBC scintigraphy with SPECT/CT both demonstrate good diagnostic accuracy for FRI”. Comment: As FDG-PET/CT becomes more available in various settings, its diagnostic performance should be specified in the text.

Performances of FDG-PET/CT were precised: “Several studies have evaluated the effectiveness of FDG-PET/CT in diagnosing FRIs, and the results are promising. FDG-PET/CT is particularly useful in complex cases where conventional imaging is inconclusive. It may guide surgical decision-making and antibiotic therapy by pinpointing active infection sites.”

-Table 3: 9th row: Comment: Although some of the new β-lactam/beta-lactamase antibiotics have shown activity against S. maltophilia, authors should clarify that, depending on susceptibility, the treatment of choice remains TMP/SMX or levofloxacin.

A note was added for S. maltophilia to the caption of table 3.

-Lines 302-303: “Pharmacokinetic studies demonstrated that aminoglycosides effectively penetrate bone and joint tissues, despite their hydrophilicity” Comment: Authors should emphasize that treatment with aminoglycosides for extended periods (as required for osteoarticular infections) is limited by its toxicity and may not be routinely recommended.

The manuscript was revised accordingly.

-Lines 315-318: “The main antibiotics available by oral route that have been used to treat BJI infections are amoxicillin, amoxicillin-clavulanate, cephalexin, flucloxacillin, fluoroquinolones, clindamycin, TMP-SMZ, tetracyclines, oxazolidinones (linezolid, tedizolid), rifampicin (and other rifamycins), and metronidazole.” Comment: A warning about bone marrow toxicity and neuropathy caused by linezolid when used for more than 2 weeks should be included.

The manuscript was revised accordingly.

-Lines 330-331: “Fosfomycin exhibits satisfactory bone concentration (ratio bone/plasma = 0.46)” Comment: The fosfomycin dose should be specified or recommended for osteoarticular infections, as many oral dosages vary depending on the site of infection.

The manuscript was revised accordingly: “In the event that no oral antibiotic treatment is available, fosfomycin IV exhibits satisfactory bone concentration (ratio bone/plasma = 0.46) [70].” (The study cited refers to IV fosfomycin)

Minor comments:

Line 228: Pseudomonas sp. Comment: Please italicize to Pseudomonas spp. (and add a second p at the end).

The manuscript was revised accordingly

Table 3: line 10: Imipeneme/relebactam should be corrected to Imipenem/relebactam

The manuscript was revised accordingly

Line 320: Pseudomonas sp.

Comment: Please italicize to Pseudomonas spp. (and add a second p at the end).

The manuscript was revised accordingly

Table 4: Comment: Many of the words contained in the table, besides the microorganisms’ names, are incorrectly italicized.

The manuscript was revised accordingly

Reviewer 3 Report

Comments and Suggestions for Authors

Introduction

This section is well-written but have to be improved by a restructuration.

This section have to follow a logical flow: 

  • Burden of disease (epidemiology & cost)

  • Pathogenesis & clinical challenges

  • Gaps in current understanding

  • Purpose of the review

    After incidence and infection rates, immediately highlight the economic burden instead of leaving cost as an isolated fact.
  • Clarify in the introduction whether this is a narrative review or a systematic/scoping review.

Methodology

Separate literature search methodology (databases, inclusion/exclusion criteria, time frame) from the results/evidence summary. For a review article, this distinction is essential.

The description of the literature search is not appropriate: only PubMed is mentioned, with English-only studies post-1980, but no PRISMA-style description of selection, number of articles screened, or exclusion criteria.

Expand on search strategy (“PubMed was searched on [date] using the following Boolean search string:…”). Report how many studies were identified, screened, included, and excluded. If this is a systematic review, a PRISMA flow diagram is mandatory.

References Appropriate 

Tables and figures good quality and appropriate. 

Author Response

Introduction

This section is well-written but must be improved by restructuring.

This section has to follow a logical flow: 

  • Burden of disease (epidemiology & cost)
  • Pathogenesis & clinical challenges
  • Gaps in current understanding
  • Purpose of the review

After incidence and infection rates, immediately highlight the economic burden instead of leaving cost as an isolated fact.

  • Clarify in the introduction whether this is a narrative review or a systematic/scoping review.

Thanks for all these relevant comments. We reshaped the introduction and included these new parts accordingly.

“Fracture fixation devices with 2 million fracture fixation devices inserted annually in the United States [1]. Fracture-related infections (FRI) are a serious complication and one of the most frequently encountered bone and joint infections, with annual projected infections of 100,000 [1][1].

FRI the overall FRI rate is estimated at 5%. Incidence ranges from 1-2% after internal fixation of closed fractures, to more than 30% after open fracture fixation [2]. This incidence is rising, representing an economic burden with substantial healthcare costs. A complex multidisciplinary management is necessary associated with prolonged hospitalization. In 2019, 178 million new fractures occurred worldwide. Economic impact is significant with total inpatient cost evaluated over a decade, up to 157million dollars in Australia. Infected trauma cases cost three times more than uninfected cases. Lancet 2024

The average cost of treatment is estimated between $15,000-25,000 [1]. Early or acute infections less than 2 weeks after implantation are the most common and are associated with virulent organisms such as Staphylococcus aureus and aerobic Gram-negative bacilli [2]. Delayed infections occur 4-10 weeks and late infections more than 10 weeks after surgery [2]. Delayed and late infections are harder to eradicate due to the presence of biofilm that shield bacteria from immunity and antibiotics. This persistence complicates eradication and favors chronicity. In contrast to prosthetic joint infections, hematogenous infection of fracture fixation devices from a distant site are uncommon [2]. 

Diagnostics can be challenging in case of subtle or delayed symptoms, requiring complex surgeries associated with prolonged antimicrobial therapy. Finally, multidisciplinary approach is crucial for optimal care and follow up of these complex cases.

Some gaps persist in our understanding of FRI, leading to difficulties in their management. The mechanism of biofilm formation, especially in vivo, remains incomplete, limiting the activity of treatments. Diagnostic tools show limitations in low grade infections delaying the management. Moreover, polymicrobial infections are frequent in open fractures, with evolving resistance profiles, challenging the choice of adapted antibiotics.

In this review, several ‘hot topics’ on FRIs were selected and reviewed by members of the Bone and Skin & Soft Tissue Infections Working Group of the International Society of Antimicrobial Chemotherapy (ISAC). This group includes international scientists, microbiology and infectious diseases clinicians, and academics whose aim is to advance the education and science of infection management. This paper aims to summarize an updated comprehensive review of current evidence, providing a summary of the various aspects of FRI: epidemiology, diagnosis, clinical signs, treatment concepts and prophylaxis. Opinions from experts on surgery, infectious diseases and microbiology are highlighted as well as areas for future study and research.”

Methodology

Separate literature search methodology (databases, inclusion/exclusion criteria, time frame) from the results/evidence summary. For a review article, this distinction is essential.The description of the literature search is not appropriate: only PubMed is mentioned, with English-only studies post-1980, but no PRISMA-style description of selection, number of articles screened, or exclusion criteria.Expand on search strategy (“PubMed was searched on [date] using the following Boolean search string…”). Report how many studies were identified, screened, included, and excluded. If this is a systematic review, a PRISMA flow diagram is mandatory.

Thanks for this comments on the methodology part. We completed it according to your suggestions. “Data was collected on the definition of infectious complications after fracture fixation used in each study. Study selection was performed first by reviewing titles and abstracts for relevance, and then by getting the full texts of relevant articles. Full-text articles were reviewed in a narrative synthesis. Experts in bone and joint infections, including infectious disease specialists, surgeons and microbiology specialists, examined data concerning fracture-related infections. Infections associated with fractures were retrieved by searching Embase, Cochrane, Google Scholar, Medline (OvidSP), PubMed publisher. Articles in English published after 1980 were considered. Keywords included fracture-related infection, orthopedic infection; osteosynthesis, open limb, internal fixation, osteomyelitis, antibiotic treatment, injury, surgical implant. In addition, when scientific evidence was lacking, recommendations were based on expert opinion.”

References Appropriate 

Tables and figures good quality and appropriate. 

Round 2

Reviewer 2 Report

Comments and Suggestions for Authors

Please note that throught the manuscript there are still many bacterial names without italicize, otheriwse, the authors answered all the queries and suggestions to my satisfaction.

Reviewer 3 Report

Comments and Suggestions for Authors

The manuscript was improved like as suggested and now is worthy to be published.